

# Error reduction and representation in stages (ERRIS) in

# hydrological modelling for ensemble streamflow forecasting

6           Ming Li[1], Q.J. Wang[2], James C. Bennett[2] and David E. Robertson[2]
7                     [1]CSIRO Data61, Floreat, WA, Australia
8                 [2]CSIRO Land and Water, Clayton, Victoria, Australia

**Corresponding Author:**
Dr Ming Li
CSIRO Data61
Private Bag 5, Wembley, WA 6014
Australia
Phone +61-8-9333 6417
Fax +61-8-9333 6121
Email Ming.Li@csiro.au





**ABSTRACT**:
This study develops a new error modelling method for short-term and real-time streamflow
forecasting, called error reduction and representation in stages (ERRIS). The novelty of ERRIS
is that it does not rely on a single complex error model but runs a sequence of simple error
models through four stages. At each stage, an error model attempts to incrementally improve
over the previous stage. Stage 1 establishes parameters of a hydrological model and parameters
of a transformation function for data normalization, Stage 2 applies a bias-correction, Stage 3
applies an autoregressive (AR) updating, and Stage 4 applies a Gaussian mixture distribution to
represent model residuals. For a range of catchments, the forecasts at the end of Stage 4 are
shown to be much more accurate than at Stage 1 and to be highly reliable in representing forecast
uncertainty. In particular, the forecasts become more accurate by applying the AR updating at
Stage 3, and more reliable in uncertainty spread by using a mixture of two Gaussian distributions
to represent the residuals at Stage 4. While the method produces ensemble forecasts, ERRIS can
be applied to any existing calibrated hydrological models, including those calibrated to
deterministic (e.g. least-squares) objectives.
**KEYWORDS**:   streamflow forecasting, updating, residual distribution, multi-stage error
modelling, ensemble forecasting



## 1.    Introduction

Streamflow forecasts have long been used to support decision making for managing river
conditions, such as flood emergency response and for optimal water allocation. Recently, much
research has been carried out on ensemble streamflow forecasting [e.g. *Alfieri et al.*, 2013;
*Bennett et al.*, 2014a; *Demargne et al.*, 2014; *Thielen et al.*, 2009], encouraged by research
communities such as the Hydrological Ensemble Prediction Experiment (HEPEX -
http://hepex.org/). In recognition that streamflow forecasts can be subject to significant errors,
forecast ensembles are used to represent forecast uncertainty. In producing ensemble forecasts,
one aims to reduce forecast uncertainty as much as possible to give the most accurate forecasts.
One also aims to represent the remaining forecast uncertainty reliably to give the right
distribution among ensemble members.
Streamflow forecasts are usually made by initializing hydrological models (e.g. conceptual
rainfall-runoff models) and then forcing them with forecast rainfall. There are a number of
sources of errors in streamflow forecasts, including errors in measurement of observed rainfall
and streamflow, errors in hydrological model structure, errors in estimated model parameters,
and errors in forecast rainfall. Ideal hydrological error quantification would account for each
individual source of errors explicitly and reliably, such that all sources of errors would
accumulate to accurately represent overall errors in the streamflow forecasts. Various attempts
have been made to identify and decompose the sources of errors, by methods such as sequential
optimization and data assimilation [*Vrugt et al.*, 2005], sequential assimilation [*Moradkhani et*
*al.*, 2005], the Bayesian total error analysis (BATEA) [*Kavetski et al.*, 2006a; b; *Kuczera et al.*,
2006], and Integrated Bayesian Uncertainty Estimator (IBUNE) [*Ajami et al.*, 2007]. Such
methods are useful for attempting to separate the major sources of errors, identifying deficiencies
of model structure, performing parameter sensitivity analyses and comparing different
hydrological models, without confounding input and output errors. However, because of a lack
of information on the different sources of errors and on how they interact with each other, it is
highly challenging to apply an error decomposition approach to arrive at statistically reliable
overall errors in streamflow forecasts [*Renard et al.*, 2010].
An alternative approach is to consider only the overall errors of forecasts, without attempting to
explain the sources of errors. An estimate of the overall error of a forecast is the residual, defined
as the difference between modelled streamflow and observations. We now concentrate our
discussion on residuals, but we will continue to refer to models of residuals as 'error models',
following common practice. Residuals of a series of forecasts form a time series. The most
traditional and simplest error model, related to the classical least squares calibration, is based on
the assumption of uncorrelated homoscedastic Gaussian residuals in the time series of residuals
[*Diskin and Simon*, 1977]. This assumption is generally not valid for hydrological applications,
where residuals are frequently auto-correlated, heteroscedastic and non-Gaussian [*Kuczera*,
1983; *Sorooshian and Dracup*, 1980]. More sophisticated error models have been developed to
address correlation, variance structure and the distribution of residuals. Autoregressive models
have been widely used to account for auto-correlation of residuals [e.g. *Bates and Campbell*,
2001; *Xiong and O'Connor*, 2002]. Heteroscedasticity may be explicitly dealt with by describing
the variance of residuals as a function of some state-dependent variables (e.g. observed
streamflow, dry/wet seasons) [e.g. *Evin et al.*, 2013; *Schaefli et al.*, 2007; *Yang et al.*, 2007].
Non-Gaussianity of residuals may be explicitly represented by non-Gaussian probability





distributions [e.g. *Marshall et al.*, 2006; *Schaefli et al.*, 2007; *Schoups and Vrugt*, 2010].
Heteroscedasticity and non-Gaussianity of residuals may also be dealt with implicitly, and often
more conveniently, by using data transformation to normalize the residuals and stabilize their
variance [e.g. *Thiemann et al.*, 2001; *Thyer et al.*, 2002; *Wang et al.*, 2012].
The approach of dealing with only the residuals, without considering the individual sources of
errors, greatly simplifies the problem of error modelling for the purpose of error reduction and
quantification. Broadly, previous attempts to model residuals can be divided into 'post-
processor' methods that separate the estimation of hydrological model parameters from the
estimation of error model parameters, and 'joint inference' methods that estimate all
parameters at once. Post-processor methods (e.g. *Evin et al.* [2014]] are often held to be less
theoretically desirable than joint inference methods [e.g. *Kuczera*, 1983*; Bates and*
*Campbell*, 2001]. This is because joint inference methods aspire to a complete description of
the behavior of errors, including behaviors that arise from interactions between parameters
from hydrological and error models [see discussion in *Evin et al.*, 2014]. Unfortunately joint
inference methods can have serious limitations for operational forecasting of streamflows.
*Li et al.* [2015] showed that a joint inference method caused poor performance in the
hydrological model when it was isolated from the error model (we will call this the 'base'
hydrological model). Error models that account for auto-correlated residuals have less
influence on forecasts as lead-time increases. Thus as lead-time increases, and the influence
of the error model decreases, the quality of the forecast relies on the performance of the
base hydrological model. *Evin et al.* [2014] demonstrated another (and perhaps more
egregious) limitation of joint inference methods: joint estimation can result in deleterious





interference between error model and hydrological model parameters, leading to poor out-
of-sample streamflow predictions. In our experience, interactions between parameters of the
hydrological model and the error model can make it very difficult to calibrate the models jointly.
The shape of the distribution of forecast residuals can change markedly after hydrological model
forecasts are updated, for example with an autoregressive error model. Despite considerable
progress in hydrological uncertainty modelling, few studies in the literature present model
forecasts (or simulations) that are practically reliable when error updating is applied [e.g. *Gragne*
*et al.*, 2015; *Schoups and Vrugt*, 2010].
This paper presents a new error modelling method, called error reduction and representation
in stages (ERRIS), for real-time and short-term streamflow forecasting applications. ERRIS
is a post-processing method developed to deal with the overall errors of streamflow
forecasts resulting from hydrological uncertainty only. Errors in streamflow forecasts due to
uncertainty in weather (precipitation in particular) forecasts are modelled separately by
using ensemble weather forecasts [*Bennett et al.*, 2014c; *Robertson et al.*, 2013; *Shrestha et*
*al.*, 2013]. For convenience, in this study we use the term *streamflow forecast* to mean one-
step-ahead model prediction of streamflow, given observed weather and streamflow up to
just before the forecast start time and assuming a one-step-ahead weather forecast that turns
out to perfectly match observations. In future work, we will extend ERRIS to multiple-step-
ahead streamflow forecasting.
The novelty of ERRIS is that it does not rely on a single complex error model, but runs a
sequence of simple error models through multiple stages. We start with a very simple model
of independent Gaussian residuals after data transformation to determine hydrological model



parameters. At each subsequent stage, an error model is introduced to improve over the
previous stage and to finalize the representation, including associated parameter values, of one
particular statistical feature (bias, correlation in residuals or a non-Gaussian distribution).
ERRIS progressively refines model features, focusing only on a small number of model
parameters at each stage. This is achieved by estimating the values for a core set of
parameters at each stage and holding them constant at subsequent stages. In doing so,
ERRIS avoids the problems associated with parameter interactions that can occur under
joint inference methods.
This paper is organized as follows. The ERRIS method is described in detail in Section 2. A
case study is introduced in Section 3. Major results are presented in Section 4, followed by
discussion and further results in Section 5. Conclusions are made in Section 6.
**2.    The error reduction and representation in stages (ERRIS) method**
**2.1.    Model formulation**
*Stage 1: Transformation and hydrological modelling*
We start from a simplified version of the seasonally invariant error model described by *Li et al.*
[2013] to calibrate the hydrological model in the ERRIS method. At stage 1, we apply the
log-sinh transformation [*Wang et al.*, 2012]
$$f(Q) = b^{-1} \log\{\sinh(a + bQ)\},$$    (1)
where $a$ and $b$ are transformation parameters, to the raw values of streamflow $Q$. We assume at
this stage that hydrological model forecast residuals are independent and, in the transformed
space, follow a Gaussian distribution with a constant variance. The log-sinh transformation
has been applied to a wide range of hydrological data [e.g. *Li et al.*, 2013; *Peng et al.*, 2014;
*Robertson et al.*, 2013; *Shrestha et al.*, 2015; *Zhao et al.*, 2015] including extreme daily
streamflow values [*Bennett et al.*, 2014b] to normalize data and stabilize variance, and has been
shown to perform at least as well as other commonly used transformations [*Del Giudice et al.*,
2013; *Wang et al.*, 2012].
We denote the observed and simulated streamflows at day $t$ by $Q(t)$ and $\tilde{Q}(t)$, respectively.
The error model at Stage 1 is mathematically specified as
$Z(t) = f(Q(t))$ (2)
$\tilde{Z}_1(t) = f(\tilde{Q}(t))$ (3)
$Z(t) \sim N\left(\tilde{Z}_1(t), \sigma_1^2\right)$ (4)
where $N$ denotes a Gaussian distribution of the model residuals in the transformed space at
Stage 1, with mean $\tilde{Z}_1(t)$ and standard deviation $\sigma_1$. We will use similar notations (e.g. $\tilde{Q}$, $Z$,
$\tilde{Z}$ and $\sigma$) for all stages in the ERRIS method, with stages distinguished by subscripts (i.e. 1,
2, 3, 4) . No autocorrelation within the forecast residuals is assumed at Stage 1. This avoids
the potential parameter interference between the autocorrelation parameter and hydrological
model parameters (e.g. parameters describing time persistence of the hydrograph) when the
hydrological model is jointly calibrated with the error model.




At the end of Stage 1, the simulated streamflow $\tilde{Q}(t)$ is taken as the forecast median of the
ensemble streamflow forecast.
*Stage 2: Linear bias correction*
At Stage 1, we assume that the hydrological simulation is overall unbiased. However, the
hydrological model often over-estimates low flows and under-estimates high flows. At Stage 2,
we adopt a simple but effective bias-correction scheme firstly introduced by *Wang et al.* [2014]
to revise the the forecast value made at Stage 1. This bias correction describes the forecast bias in
the transformed domain by a linear function. Because the bias-correction is applied to
transformed data, it is able to cope with conditional biases (biases that vary with flow magnitude)
that are often present in hydrological model simulations, even if these vary in a strongly non-
linear way. We express the specific error model structure of Stage 2 as
$$\tilde{Z}_2(t) = c + d\tilde{Z}_1(t) \tag{5}$$
$$Z(t) \sim N\left(\tilde{Z}_2(t), \sigma_2^2\right) \tag{6}$$
where $c$ and $d$ represent the intercept and slope parameters of the bias correction and $\sigma_2$
denotes the standard deviation of the residuals at Stage 2. The slope parameter $d$ allows much
flexibility in the bias correction. When $d$ equals 1, this bias correction becomes a simple
additive correction. When $d$ equals 0, the bias-correction forces the forecast to approach a
constant (in additional to uncertainty). This may happen when the hydrological forecast performs
worse than climatology (i.e. long-term average). When $d$ is greater than 1, the bias-correction


can correct the very strongly conditional biases, as might be found in ephemeral and intermittent
catchments.
At the end of Stage 2, the forecast median in the orginal space is revised to
$\tilde{Q}_2(t) = f^{-1}\left(\tilde{Z}_2(t)\right),$                          (7)
where $f^{-1}(x) = b^{-1}\operatorname{arsinh}\{\exp(bx) - a\}$ is the back-transformation of the log-sinh transformation
given in Equation (1).
*Stage 3: AR updating*
At Stage 3, we no longer assume that forecast residuals are independent, and use an AR-
based error model to describe the correlation structure of forecast residuals. The AR-based
error model enables the ERRIS method to correct forecast residuals based on the latest
available observations of streamflow. Specifically, we assume that the forecast residuals at
Stage 2 follow a restricted AR error model described by *Li et al.* [2015]. The error model at
Stage 3 can be written as

$$\tilde{Z}_3(t) = \begin{cases} \tilde{Z}_2(t) + \rho\left(Z(t-1) - \tilde{Z}_2(t-1)\right) & \text{if } \left|\tilde{Q}_3^*(t) - \tilde{Q}_2(t)\right| \le \left|Q(t-1) - \tilde{Q}_2(t-1)\right| \\ f\left(\tilde{Q}_2(t) + Q(t-1) - \tilde{Q}_2(t-1)\right) & \text{otherwise} \end{cases}$$

                                     (8)

$Z(t) \sim N\left(\tilde{Z}_3(t), \sigma_3^2\right)$                          (9)
where $\tilde{Q}_3^*(t) = f^{-1}\left(\tilde{Z}_2(t) + \rho\left(Z(t-1) - \tilde{Z}_2(t-1)\right)\right)$ is the updated streamflow without applying
the restriction, and $\rho$ and $\sigma_3$ are the lag-1 autocorrelation parameter and the standard deviation





of the residuals at Stage 3, respectively. *Li et al.* [2015] demonstrated that when AR models are
applied to normalized residuals without restriction, over-correction of forecasts can occur,
particularly at the peak or on the rise of a hydrograph. Equation (8) uses the restricted AR error
model to reduce the tendency to over-correct forecasts. In Equation (8) the forecast median,
denoted by $\tilde{Q}_3(t)$, is given by
$$\tilde{Q}_3(t) = \begin{cases} \tilde{Q}_3^*(t) & \text{if } \left| \tilde{Q}_3^*(t) - \tilde{Q}_2(t) \right| \le \left| Q(t-1) - \tilde{Q}_2(t-1) \right| \\ \tilde{Q}_2(t) + Q(t-1) - \tilde{Q}_2(t-1) & \text{otherwise} \end{cases}. \qquad (10)$$
The forecast at Stage 3 updates $\tilde{Q}_2(t)$ based on the latest observed streamflow $Q(t-1)$ and its
difference from $\tilde{Q}_2(t-1)$. Therefore, more information (i.e. streamflow observations at the
previous time step) is required to generate streamflow forecasts at Stage 3 than at the previous
two stages.
*Stage 4: Residual distribution refinement*
In Section 4, we will demonstrate that the residuals after Stages 1 and 2 are well described
by Gaussian distributions, but the shape of the residual distribution after Stage 3
dramatically changes. In particular, the distribution of the residuals after Stage 3 looks more
peaked and has longer tails than a Gaussian distribution. The reason for the non-Gaussian
residuals after Stage 3 is as follows. The AR updating at Stage 3 is very effective in
correcting small residuals especially at hydrograph recession and therefore reducing
residuals to very small values. The updating, however, is not very effective around peaks,





where the residuals remain large even in the transformed space. This results in a centrally
peaked and long tailed distribution of residuals after Stage 3.
At Stage 4, we use a non-Gaussian distribution to describe the model residuals from Stage 3.
Several long-tailed distributions have been used in hydrological modelling studies, such as
the finite mixture distribution [*Schaefli et al.*, 2007; *Smith et al.*, 2010], the exponential
power distribution [*Schoups and Vrugt*, 2010] and Student's t-distribution [*Marshall et al.*,
2006]. In this study, we assume that the model residuals can be grouped into two categories
with respect to variance and thus choose a two-component Gaussian mixture distribution. It is
possible to use more than two components, but we will show in our case study that two
components are sufficient. We discuss the possibility of using other long-tailed distributions
in Section 5.1.
Using a two-component Gaussian mixture distribution, we express the residual model at
Stage 4 as
$$\tilde{Z}_4(t) = \tilde{Z}_3(t) \tag{11}$$
$$Z(t) \sim MN\left(\tilde{Z}_4(t), \sigma_{4,1}^2, \sigma_{4,2}^2, w\right), \tag{12}$$
where $MN\left(\tilde{Z}_4(t), \sigma_{4,1}^2, \sigma_{4,2}^2, p\right)$ represents a mixture of two Gaussian distributions $N\left(\tilde{Z}_4(t), \sigma_{4,1}^2\right)$
and $N\left(\tilde{Z}_4(t), \sigma_{4,2}^2\right)$ with weights $W$ and $1-w$. The corresponding probability density function
of $MN\left(\tilde{Z}_4(t), \sigma_{4,1}^2, \sigma_{4,2}^2, w\right)$, denoted by $pdf\left(Z(t) \mid \tilde{Z}_4(t), \sigma_{4,1}^2, \sigma_{4,2}^2, w\right)$, can be explicitly written as a
weighted sum of two Gaussian probability density functions





$$pdf\left(Z(t)\,|\,\tilde{Z}_4(t), \sigma_{4,1}^2, \sigma_{4,2}^2, w\right) = w\phi\left(Z(t)\,|\,\tilde{Z}_4(t), \sigma_{4,1}^2\right) + (1-w)\phi\left(Z(t)\,|\,\tilde{Z}_4(t), \sigma_{4,2}^2\right). \qquad (13)$$
where $\phi$ is the probability density function (PDF) of a Gaussian distribution. We assume that
$\sigma_{4,1} < \sigma_{4,2}$ to make the two components identifiable. This assumption implies that $w$ represents
the probability associated with the mixture component that has a smaller variance.
The four stages of the ERRIS method are summarized in Table 1.
**2.2.     Model estimation**
The maximum likelihood estimation [*Li et al.*, 2013; *Wang et al.*, 2009] is used to estimate
model parameters at all four stages. Denote the parameter set as $\theta_S$ for Stage *S*. The likelihood
functions for the four stages are given by
$$L_S\left(\theta_S\right) = \prod_t J_{z \to Q}\phi\left(Z(t)\,|\,\tilde{Z}_S(t), \sigma_S^2\right) \qquad (14)$$
for $S = 1, 2, 3$, and
$$L_4\left(\theta_4\right) = \prod_t J_{z \to Q}\,pdf\left(Z(t)\,|\,\tilde{Z}_4(t), \sigma_{4,1}^2, \sigma_{4,2}^2, w\right) \qquad (15)$$
where $J_{z \to Q} = 1/\tanh\{a + bQ(t)\}$ is the Jacobian determinant of the log-sinh transformation.
At Stage 1, the hydrological model parameters, transformation parameters ($a$ and $b$) and the
residual standard deviation ($\sigma_1$) are jointly estimated by maximizing the likelihood function. It
is also possible to use a set of parameters already calibrated for the hydrological model (using a





different objective, such as the least sum of squared errors) and estimate at Stage 1 only the
transformation parameters and the residual standard deviation (see discussion in Section 5.2). At
the end of Stage 1, the values of the hydrological parameters and the transformation parameters
are concluded, without further changes in subsequent stages.
At Stage 2, the bias correction parameters ($c$ and $d$) and the residual standard deviation ($\sigma_2$)
are estimated by maximizing the likelihood function. At the end of Stage 2, the values of the bias
correction parameters are concluded. At Stage 3, the auto-correlation coefficient ($\rho$) and the
residual standard deviation ($\sigma_3$) are estimated. At the end of Stage 3, the value of the auto-
correlation coefficient is concluded. At Stage 4, the model residual parameters ($\sigma_{4,1}, \sigma_{4,2}$ and
$W$) are finalized. Note that parameters $\sigma_1$, $\sigma_2$ and $\sigma_3$ are only intermediate parameters to assist
in the estimation of other parameters at corresponding stages.
The Shuffled Complex Evolution (SCE) algorithm [*Duan et al.*, 1994] is used to maximize the
log likelihood function at Stage 1, where a number of parameters are required to be calibrated.
The Simplex algorithm [*Nelder and Mead*, 1965] is used in the likelihood-based calibration at
other stages, where fewer parameters are present. We use different optimization algorithms
because the Simplex algorithm is more computationally efficient when the number of parameters
is small.



### 277    2.3.    Model verification

We use several performance measures to evaluate the ensemble forecasts derived at each
stage. The evaluation criteria suggested by *Engeland et al.* [2010] are used to test for
important attributes of ensemble forecasts including *reliability*, *sharpness* and *efficiency*.
*Reliability* is often described as the property of statistical consistency, which allows
ensemble forecasts to reproduce the frequency of an event. Reliability can be checked by the
forecast probability integral transform (PIT) of streamflow observations, defined by
$$\pi_t = F_t\big(Q(t)\big) \tag{15}$$
where $F_t$ is the forecast CDF of the streamflow at time $t$. In the case of zero flows, we use the
pseudo PIT [*Wang and Robertson*, 2011], which is randomly generated from a uniform
distribution with a range $[0, \pi_t]$. If a forecast is reliable, $\pi_t$ follows a uniform distribution over
[0,1]. We graphically examine $\pi_t$ with the corresponding theoretical quantile of the uniform
distribution. A perfectly reliable forecast follows the 1:1 line. In addition, PIT diagrams can be
summarized by the $\alpha$-index [*Renard et al.*, 2010], defined by
$$\alpha = 1 - \frac{2}{n}\sum_{t=1}^{n}\left|\pi_t^* - \frac{t}{n+1}\right|, \tag{16}$$
where $\pi_t^*$ is the sorted $\pi_t$ in increasing order. The $\alpha$-index represents the total deviation of
$\pi_t^*$ from the corresponding uniform quantile (i.e., the tendency to deviate from the bisector in
PIT diagrams). The range of the $\alpha$-index is from 0 (worst reliability) to 1 (perfect reliability).



*Sharpness* is a measure of the spread of the forecast probability distribution. Sharp forecasts
with narrow forecast intervals are often preferred by forecast users as they reduce the range
of possible outcomes that are anticipated – that is, it is easier to make decisions with sharp
forecasts. However, if a sharp forecast is unreliable, it is underconfident and is likely to lead
to poor decisions. Thus sharp forecasts are desirable, but only if the forecasts are also
reliable. We use the average width of the 95% forecast intervals (AWCI) to indicate forecast
sharpness. Wider forecast intervals suggest less sharp forecasts. In order to compare the
sharpness across different catchments, we define a score relative AWCI with respect to a
reference forecast
$$\text{Relative AWCI} = \frac{AWCI_{REF} - AWCI}{AWCI_{REF}},\qquad(17)$$
where $AWCI_{REF}$ is AWCI calculated from the reference forecast. The reference forecast in this
study is generated by resampling historical streamflows. To issue a reference forecast for a given
month/year (e.g. February 1999), we randomly draw a sample of 1000 daily streamflows that
occur in that month (e.g. February) from other years (e.g. years other than 1999) with
replacement. The relative AWCI is unitless and the maximum is one, corresponding to the
sharpest forecast.
The *Efficiency* (or accuracy) of a forecast is commonly used to assess deterministic (single-
valued) forecasts. For the ensemble forecasts we generate here, we measure the efficiency
with the well-known Nash-Sutcliffe efficiency (NSE) [*Nash and Sutcliffe*, 1970], calculated
for the forecast mean. A greater value of NSE indicates a more accurate forecast mean and thus





better forecast efficiency. We also use relative bias to assess how the forecast mean deviates
from observations.
We evaluate the overall forecast skill with a skill score derived from the widely used continuous
ranked probability score (CRPS) [*Gneiting and Katzfuss*, 2014; *Grimit et al.*, 2006; *Wang et*
*al.*, 2009] (denoted by $CRPS\_SS$). CRPS is a negatively oriented score: a smaller value of
CRPS indicates a better forecast. As with the relative AWCI, the skill score $CRPS\_SS$ is
defined as the normalized version of CRPS with respect to a reference forecast
$$CRPS\_SS = \frac{CRPS_{REF} - CRPS}{CRPS_{REF}},\qquad(18)$$
where $CRPS_{REF}$ is CRPS calculated from the reference forecast (already defined for Equation
(18), above). The maximum of $CRPS\_SS$ is 1, corresponding to a perfectly skillful forecast.
**3.    Case Study**
**3.1    Study region and data**
We select six catchments in southeast Australia and three catchments in the United States
(US) for this study (Figure 1), from a range of climatic and hydrological conditions. The
streamflow data for the Australian catchments are obtained from the Catchment Water Yield
Estimation Tool (CWYET) dataset [*Vaze et al.*, 2011]. The rainfall and potential
evaporation data for the Australian catchments are taken from the Australian Water
Availability Project (AWAP) dataset [*Jones et al.*, 2009]. All data for the US catchments are
taken from the Model Intercomparison Experiment (MOPEX) dataset [*Duan et al.*, 2006].




The Abercrombie and Emu catchments have many instances of zero flow (Table 2), and
accurate streamflow forecasting is particularly challenging for such dry catchments.
$AWCI_{REF}$ and $CRPS_{REF}$ for each catchment is given by Table 3.
**3.2    Cross-validation**
Daily streamflow is simulated with the GR4J rainfall-runoff model [*Perrin et al.*, 2003] and
then forecasted with ERRIS as described in Section 3. Forecasts are generated from
"perfect" (observed) deterministic rainfall forecasts at a lead time of one day (i.e., one time
step ahead). All results reported in this study are based on cross-validation unless specified.
Cross-validation allows us to generalize the forecast skill to data outside the sample period.
Because of data availability, we choose different study periods for Australian and US
catchments. For Australian catchments, data from 1990 to 1991 are used to warm up the
hydrological model and the data from 1992-2005 are used to generate a leave-two-years-out
cross-validation (i.e. effectively 14-fold cross-validation). For a particular year, we remove
the streamflow data from this year and the following year and apply ERRIS to forecast the
streamflow for the year. The removal of the data from the following year aims to minimize
the impact of streamflow memory on model performance. For US catchments, the data from
1979 to 1980 are used in the warm-up period and the data from 1981 to 1998 are used for a
leave-two-years-out cross-validation (i.e. effectively 18-fold cross-validation).
**4.    Results**
Figure 2 compares forecasts at different stages for an example period. In this example, we
generate daily streamflow forecasts for the Mitta Mitta catchment in the period between



01/07/2000 to 31/12/2000. The forecast mean and the 95% forecast interval are plotted against
observations. The forecast at Stage 1 (the base hydrological model forecast) frequently over-
estimates low flows, such as in the period between July and September. For high flow periods
(e.g. October), the forecast mean is generally more accurate but virtually all observations lie
within the 95% forecast intervals, suggesting that the forecast intervals are perhaps too wide (i.e.,
the forecasts may be underconfident). The forecast mean at Stage 2 is closer to the observations
and the 95% forecast intervals tend to be narrower. Stage 2 tends to overestimate high flows less
than Stage 1, but introduces the problem of underestimating high flows in some instances (e.g.
September).
The AR error updating applied in Stage 3 significantly reduces the forecast residuals, as we
expect given that streamflows are often heavily autocorrelated. The forecasts at Stage 3 are not
only more accurate but also more certain, indicated by the considerably narrower 95% forecast
intervals. The differences between Stage 3 and Stage 4 are not evident in the time-series plots, in
essence because Stage 4 is an attempt to address issues of reliability, which is difficult to see
when forecast intervals are so narrow. We give a detailed view of changes to reliability at each
stage below.
Figure 3 summarizes the performance at each stage, and generally confirms the improvements in
performance at each stage observed in Figure 2. In general, Stage 1 and Stage 2 are similarly
efficient (Figure 3b), skillful (Figure 3c), sharp (Figure 3d) and reliable (Figure 3e). As we
expect, Stage 2 forecasts are consistently less biased than Stage 1 (Figure 3a) (except for the
Hope catchment, where many instances of zero flow occur; see Table 2). Stage 3 is generally
much more efficient and skillful than Stage 1 and Stage 2. A partial exception to this is the
Abercrombie catchment, which is less efficient at Stage 3 than Stage 2. As an intermittent





catchment, the Abercrombie catchment experiences low (to zero) flows, but is also punctuated
by abrupt high flows. Stage 3 is based on the time persistence of the residuals and may introduce
more errors when flows change abruptly, which sometimes occurs in the Abercrombie
catchment.  In addition, residuals tend to be larger at higher flows and because NSE is a measure
of squared residuals, it tends to give more weights to residuals at high flows. This causes the
Abercrombie Stage 3 forecasts to be less efficient than those of Stage 2.
As we expect, Stage 3 forecasts are notably sharper than those at Stage 2 (Figure 3d). However,
this sharpness is not supported by reliability: Stage 3 forecasts tend to be much less reliable than
all other stages (Figure 3e). Figure 4 illustrates the reliability of the forecasts at each stage in
more detail with the PIT plots. The PIT plots show that the forecasts at the first two stages are
reliable (as with the $\alpha$-index in Figure 3e). However, for Stage 3 the points on the PIT plots
deviate substantially from the 1:1 line, with a clear S-shape pattern for almost all catchments (the
exception is the Tarwin catchment). A traditional interpretation of this S-shape is that the
forecasts are underconfident [*Laio and Tamea,* 2007]. However, in this case, the S-shape is
caused by the high level of kurtosis in the distribution of the residuals, as we will show below.
The $\alpha$-index from Stage 3 is smaller than those from stages 1 and 2 (the Tarwin catchment is the
only exception), confirming the lack of the reliability at Stage 3. Stage 4 consistently improves
the reliability of the forecast after the AR updating. The PIT plot at Stage 4 is much closer to the
1:1 line than that at Stage 3 and this is reflected by the $\alpha$-index, which increases for all
catchments. Stage 4 corrects the underconfident forecasts from Stage 3 and slightly decreases the
sharpness from Stage 3 (Figure 3d).





At Stage 3, unreliable forecasts are caused by representing the model residual by an
inappropriate (Gaussian) probability distribution. We compare the underlying density of the
model residuals at Stage 3, $\varepsilon(t) = Z_3(t) - \tilde{Z}_3(t)$ (fitted by the nonparametric density estimation),
with the fitted parametric densities for different distributions in Figure 5. The fitted Gaussian
density is flatter than the underlying density of $\varepsilon(t)$ in order to match the tails for each
catchment. This suggests that the residual distribution is more peaked and has longer tails than
the Gaussian distribution. As we have seen above, forecast residuals are, in general, dramatically
reduced by the AR error updating. Unfortunately, this reduction in residual does not occur at all
events, especially where abrupt changes in flow occur (and hence the assumption of strong
autocorrelation breaks down). Thus the magnitude of the forecast residuals at Stage 3 for a small
proportion of events is large relative to the majority of events. As we have seen, the practical
implication of the dichotomous behavior of the residuals is that their distribution is still bell-
shaped and symmetric but has a much longer tail than the Gaussian distribution. The Gaussian
mixture distribution treats the entire model residuals as two groups with different variances. The
Gaussian mixture distribution is able to capture the peak and tails of the underlying residual
density for all catchments, resulting in reliable ensemble forecasts that also have a highly
accurate forecast mean. As we note in the introduction, however, other distributions have also
been used to describe "peaky" data, and we explore these in the next section.
To provide a basis for any future comparisons with this study, we include example parameter
values for each stage in Table 4 (derived by calibrating each stage to the full set of data – i.e.
without cross-validation). We note that: 1) the variance parameter at Stage 3 is always much
smaller than at Stage 1 and Stage 2, which leads to the dramatic reduction in the width of





forecast intervals at this stage; and 2) that the $w$ parameter that weights the component of the
Gaussian mixture distribution with smaller variance is always greater than 0.5, confirming that
the majority of residuals take a narrow range of values as we have described.
**5.    Further results**
**5.1    Testing an alternative residual distribution**
It is possible to use long-tailed distributions other than the Gaussian mixture distribution at Stage
4. For example, Student's t-distribution is a simple long-tailed distribution that has been used in
hydrological modelling [e.g. *Marshall et al.*, 2006]. In this section we investigate whether
Student's t-distribution is a viable alternative to the Gaussian mixture distribution at Stage 4. To
do this, we modify the model residual in Equation (12) as follows
$Z(t) = \tilde{Z}_4(t) + r\xi(t)$,                                             (19)
Where $\xi(t)$ is assumed to independently follow a Student's t-distribution with $\nu$ degrees of
freedom, and $r$ is a scale parameter describing the spread and variation of the model residuals.
We first examine how well Student's t-distribution can fit the residual distribution at Stage 4 for
all nine catchments (Figure 5). High peaks and long tails of the residual densities can be captured
reasonably well by Student's t-distribution for nearly all catchments. The fitted densities of
Student's t-distribution appear more "peaked" for most catchments than those of the Gaussian
mixture distribution, which is originally used at Stage 4. Figure 6 further investigates how
Student's t-distribution can fit the upper quantile of the model residuals. There is a clear
tendency of Student's t-distribution to overestimate the upper quantile (e.g. 98% or higher) of the



model residuals (especially for the Australian catchments). These upper quantiles are more
accurately estimated by the Gaussian mixture distribution. This implies that Student's t-
distribution often has tails that are too long. We note, however, that if the ERRIS method is
tested on other catchments, it is possible that Student's t-distribution may describe the residuals
better than the Gaussian mixture distribution in some cases.
However, the very long tail of Student's t distribution can be problematic for operational
forecasting. The degrees of freedom, $\nu$, determines how heavy the tails of Student's t-
distribution are. Table 5 presents the two calibrated parameters (i.e. $\nu$ and $r$) for all catchments.
Calibrated $\nu$ values are less than 2 for eight out of nine catchments. The exception is the Hope
catchment, and even here the calibrated $\nu$ is very close to 2. It is well know that for  degrees of
freedom less than 2, Student's t-distribution is so heavy-tailed that the variance is infinite (if
$1 < \nu \leq 2$) or even undefined (if $\nu \leq 1$). This is obviously undesirable for operational forecasting:
it can cause a few forecast ensemble members to be so large that the forecast mean becomes
implausibly large. Figure 7 compares the forecast mean with observations if the model residual is
revised as Equation (19). In all catchments, in some cases forecast mean values are
unrealistically large even as observations are relatively small. Student's t-distribution is thus
prone to be too long-tailed to be practically implemented. Therefore, we do not recommend
using Student's t-distribution to describe the residual distribution at Stage 4, and advocate the
Gaussian mixture distribution as a practical alternative.
**5.2     Testing an alternatively calibrated hydrological model**
In this study, we apply a likelihood-based calibration at Stage 1 to derive the distribution of the
forecast residuals. However, in operational practice forecasters may prefer to use their own





methods for calibrating hydrological models (or it may be onerous to recalibrate large numbers
of hydrological models, whatever method is used). It is possible to simply 'bolt on' the ERRIS
method to existing hydrological models. We simply need to calibrate the transformation
parameters and the model residual standard deviation at Stage 1 while fixing the hydrological
parameters to those already calibrated. We demonstrate this by first calibrating hydrological
models with a simple least-squares objective. We then apply the ERRIS method and repeat the
cross-validation analysis.
Figure 8, an analog to Figure 3, summarizes forecast performance when the hydrological model
is calibrated to a least-squares objective. The least-squares calibration essentially maximizes
NSE as an objective, but the corresponding cross-validated NSE is not necessarily always greater
than that of the likelihood-based calibration. The forecast performance from the two different
calibrations can differ markedly at Stage 1, but is largely similar after the AR error updating at
Stage 3 and Stage 4. Thus ERRIS is flexible enough to accommodate existing hydrological
models.
Figure 9, an analog to Figure 4, compares the PIT plots for different catchments when the
hydrological model is least-squares calibrated. The main change is that the forecasts at Stage 1
are no longer reliable in many instances. This is caused by the least-squares calibration, which
does not ensure the forecast residuals are Gaussian (even after the log-sinh transformation). The
PIT plots derived from Stage 2 and Stage 3 in Figure 9 show a very similar pattern to their
counterparts in Figure 4. It suggests that poor reliability at Stage 3 occurs irrespective of the
calibration strategy employed for the hydrological model. As with Figure 4, Figure 9 shows the





Gaussian mixture distribution used at Stage 4 effectively ameliorates the problems with the
reliability of Stage 3.
**6.      Discussion**
There are several advantages of using a multi-stage error model compared to a single complex
error model. (1) The parameter estimation in ERRIS is relatively simple, and hence
computationally efficient. Only a small number of parameters are estimated at each stage. Joint
parameter estimations associated with a single complicated error model are often more
computationally demanding. (2) Interference between parameters is minimized. The parameters
of a single complex model can confound each other and the contribution of one parameter can
sometimes be explained by others. For example, the hydrological model parameters describing
soil moisture storage capacity may interfere strongly with the error parameters describing bias.
Interference between parameters can make the parameter estimation unstable, because more than
one set of parameters can achieve a similar objective function value, and thus over-fit
parameters. (3) In operational forecasting it is often important that individual components of the
forecasting model can function independently. For example, if forecasts are issued to long lead
times, the influence of an AR model diminishes as lead time extends. Thus forecasts at long lead
times rely strongly on the hydrological model (and, in our case, with a bias-correction) to be
plausible. If all parameters are estimated jointly, it is difficult to guarantee that each component
of a forecasting model can operate independently. In addition, because stages are independent, it
is possible to change a stage without affecting other stages, making the ERRIS approach easy to
extend or modify.



This paper is aimed at developing a staged error model suitable for eventual use in an operational
ensemble forecasting system. We have focused on presenting the theoretical underpinnings of
this approach, and have limited its testing to forecasting with 'perfect' (observed) rainfall
forecasts at a lead time of one day. Operational systems routinely forecast to long lead times, and
use uncertain rainfall forecasts to force hydrological models. In future work we will extend the
validation of this model to forecast multiple lead times, and couple the ERRIS approach with
reliable ensemble rainfall forecasts [*Robertson et al.*, 2013; *Shrestha et al.*, 2015].
**7.    Summary and conclusions**
In this study, we introduce the error reduction and representation in stages (ERRIS) method to
update errors and quantify uncertainty in streamflow forecasts. The first stage of ERRIS employs
a simple error model that assumes independent Gaussian residuals after the log-sinh
transformation. The second stage applies a bias-correction that is able to correct conditional and
unconditional biases, including the sometimes strongly non-linear biases that occur in
intermittent catchments. The third stage exploits autocorrelation in residuals with an AR model
to dramatically reduce forecast residuals, but this results in unreliable ensemble forecasts. In the
fourth stage a Gaussian mixture distribution is used to describe the residuals, resulting in
ensemble forecasts that are both highly accurate and very reliable. Based on extensive validation
of ERRIS, the accuracy of the forecast mean is slightly improved by the bias correction at Stage
2 and is considerably improved by the updating at Stage 3. The reliability of the forecasts at
Stage 3 becomes a problem, because the shape of the residual distribution dramatically changes.
The revision of the residual distribution at Stage 4 is effective for representing non-Gaussian
residuals and leading to highly reliable forecasts. The Gaussian mixture distribution is showed to



be more suitable than the Student's t distribution for describing the residuals after updating. We
also confirm that ERRIS is flexible enough to adapt to existing calibrated hydrological models.
**Acknowledgements**
This research has been supported by the Water Information Research and Development Alliance
(WIRADA) between the Bureau of Meteorology and CSIRO Land & Water Flagship. We would
like to thank Andrew Schepen for valuable suggestions to improve quality of the manuscript.



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





**Figure 1: Map of the catchments used in this study**




**Figure 2: An example of streamflow time-series plots for the Mitta Mitta catchment in the period between**
**01/07/2000 and 31/12/2000.**





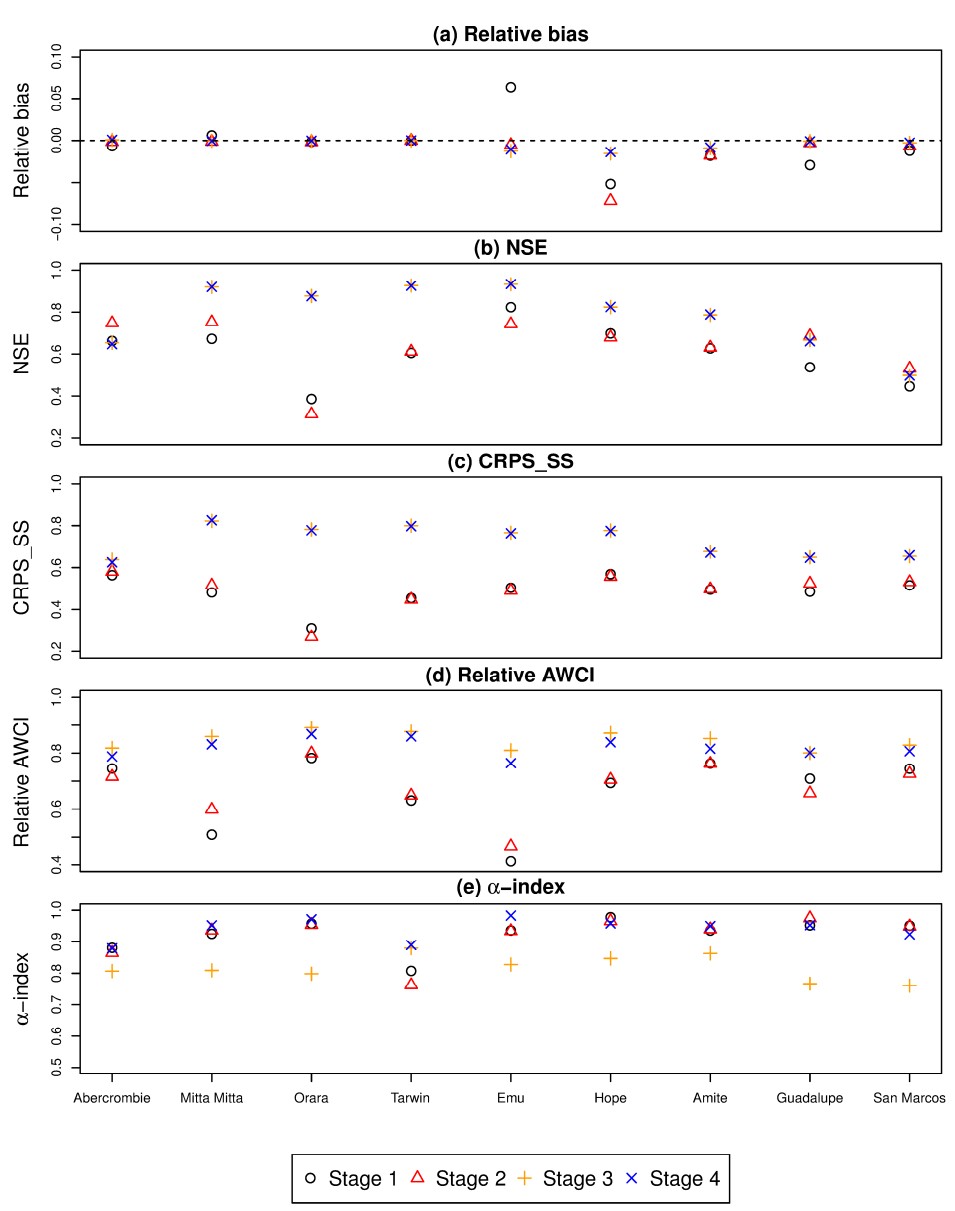


**Figure 3: Comparison of performance metrics for each catchment and each stage**





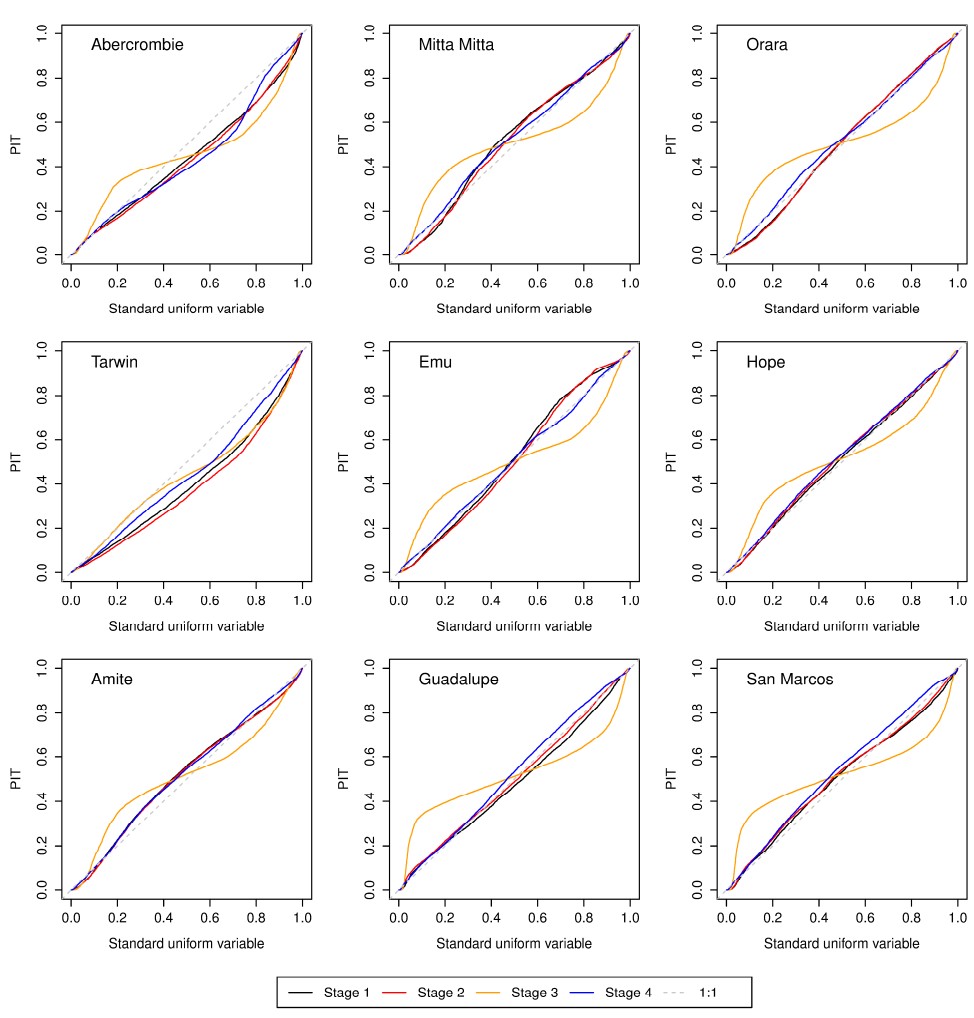

**Figure 4: Comparison of the cumulative probability distribution of the PIT at different stages.**





Page 41

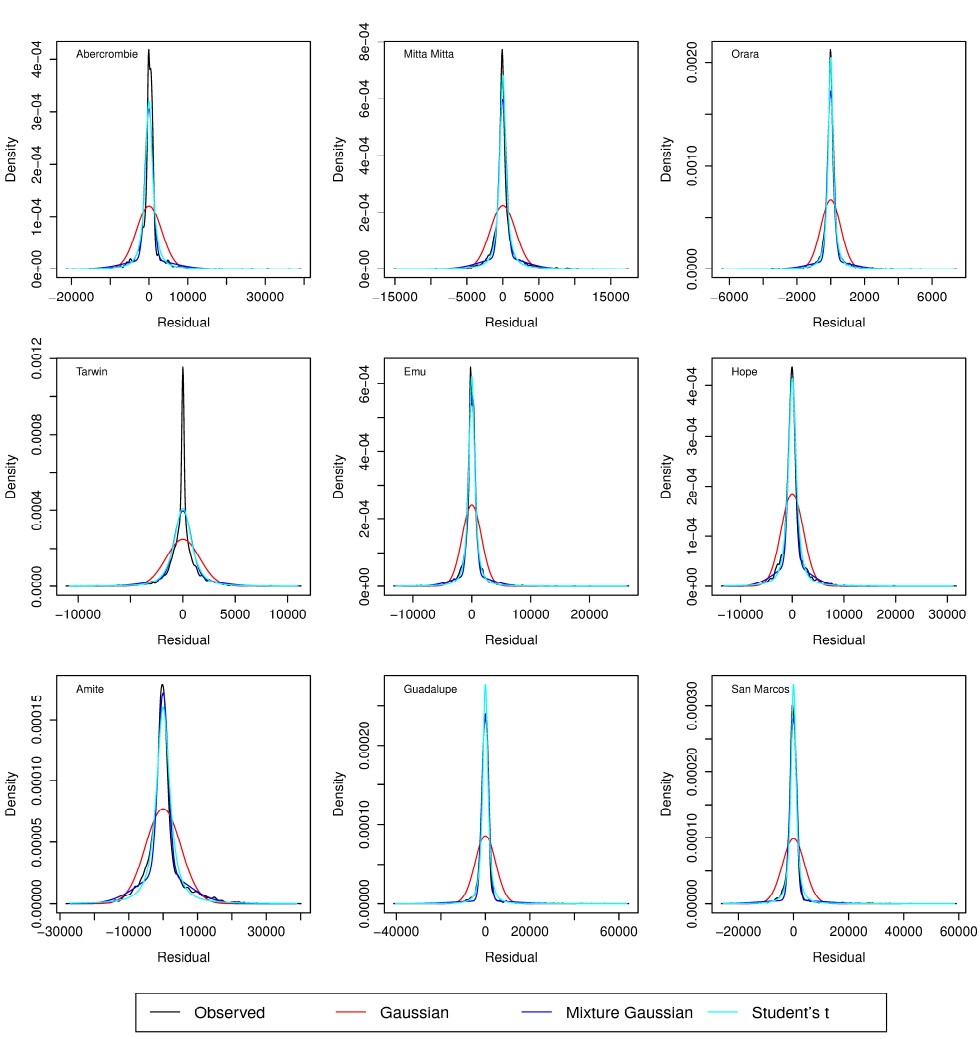

**Figure 5: Comparison of the different probability density functions fitted to the model residuals at Stage 3 for each catchment.**







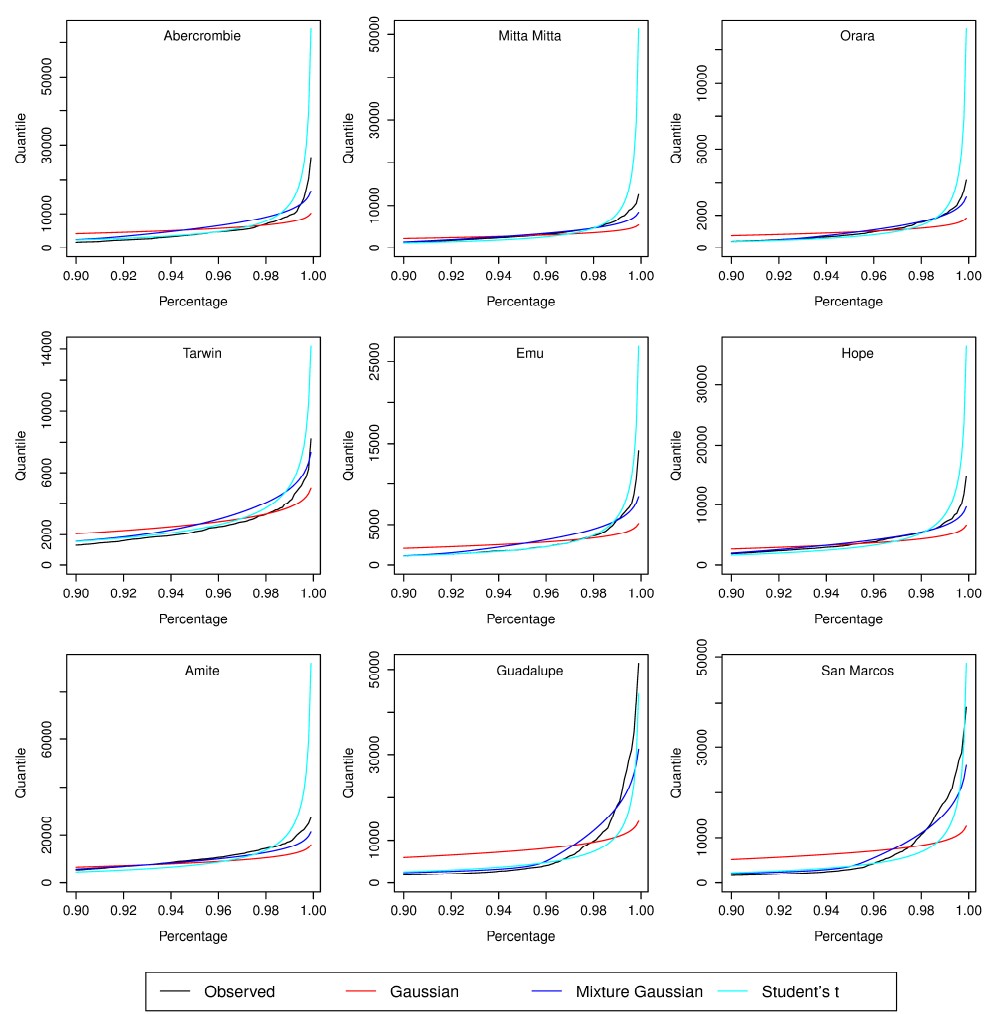

**Figure 6: Comparison of the upper quantile of the model residuals fitted by different distributions for each**
**catchment.**




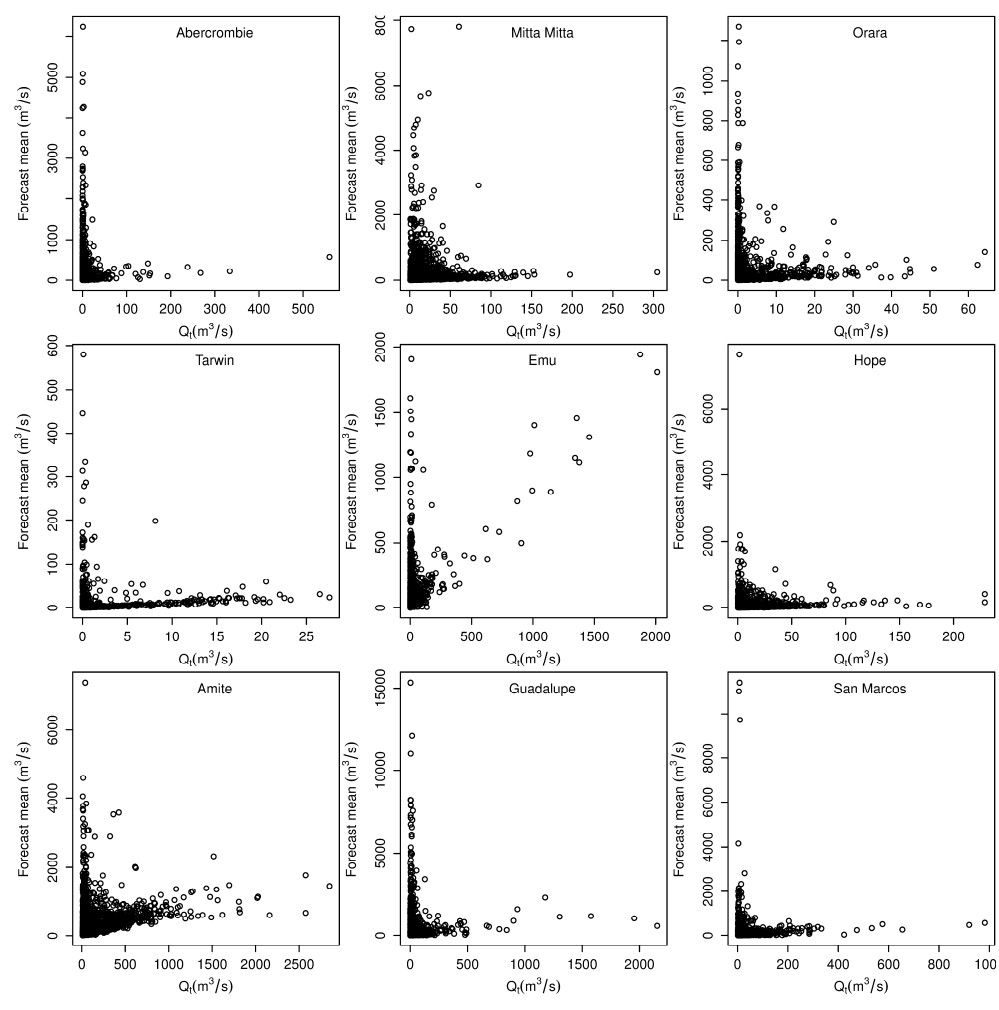

**Figure 7: Comparison of streamflow observations with streamflow forecast mean for each catchment when**
**the residual distribution is fitted by Student's t-distribution.**





Page 44

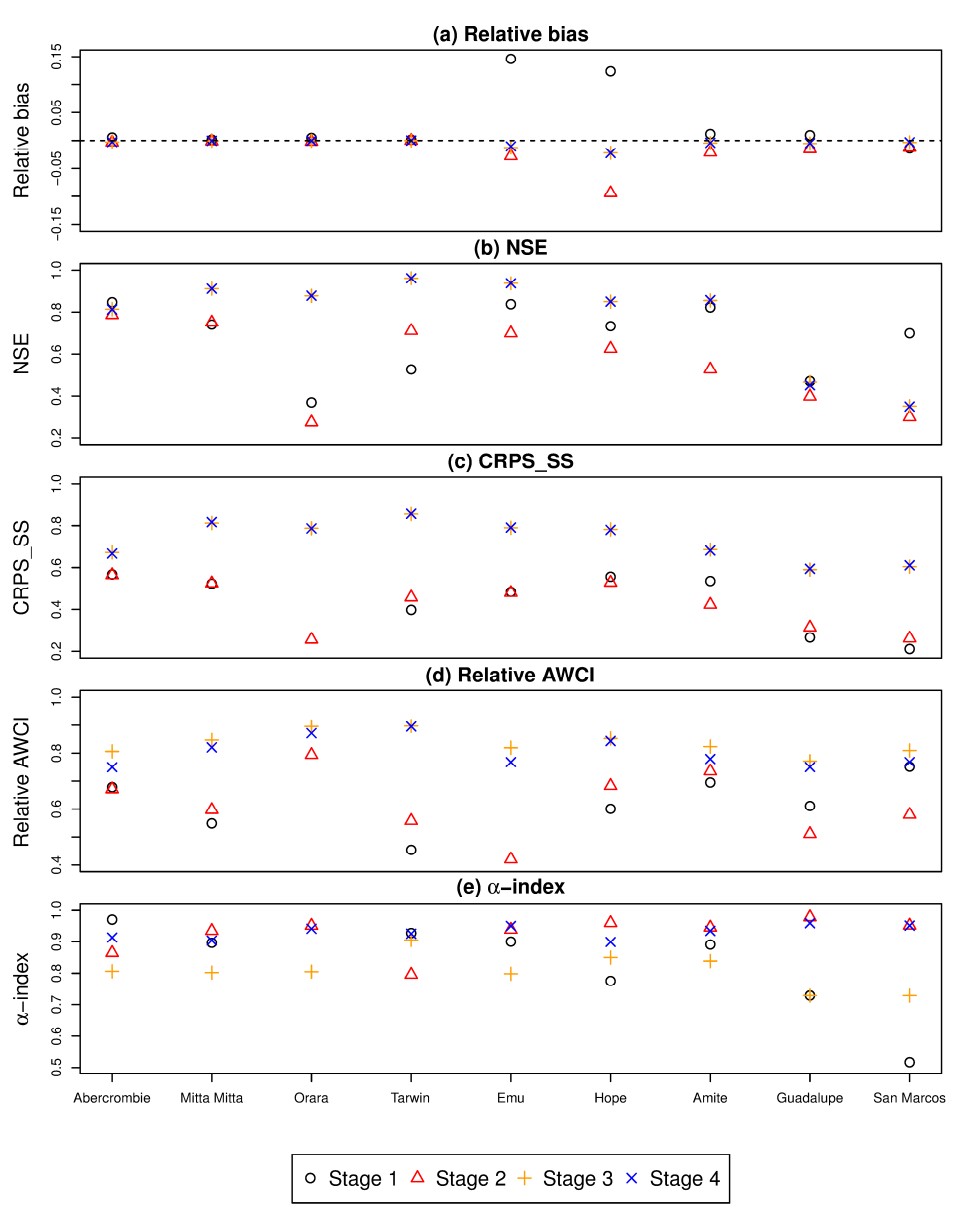


**Figure 8: Same as Figure 3 but the hydrological model is calibrated by the least-squares method.**


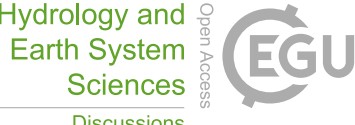




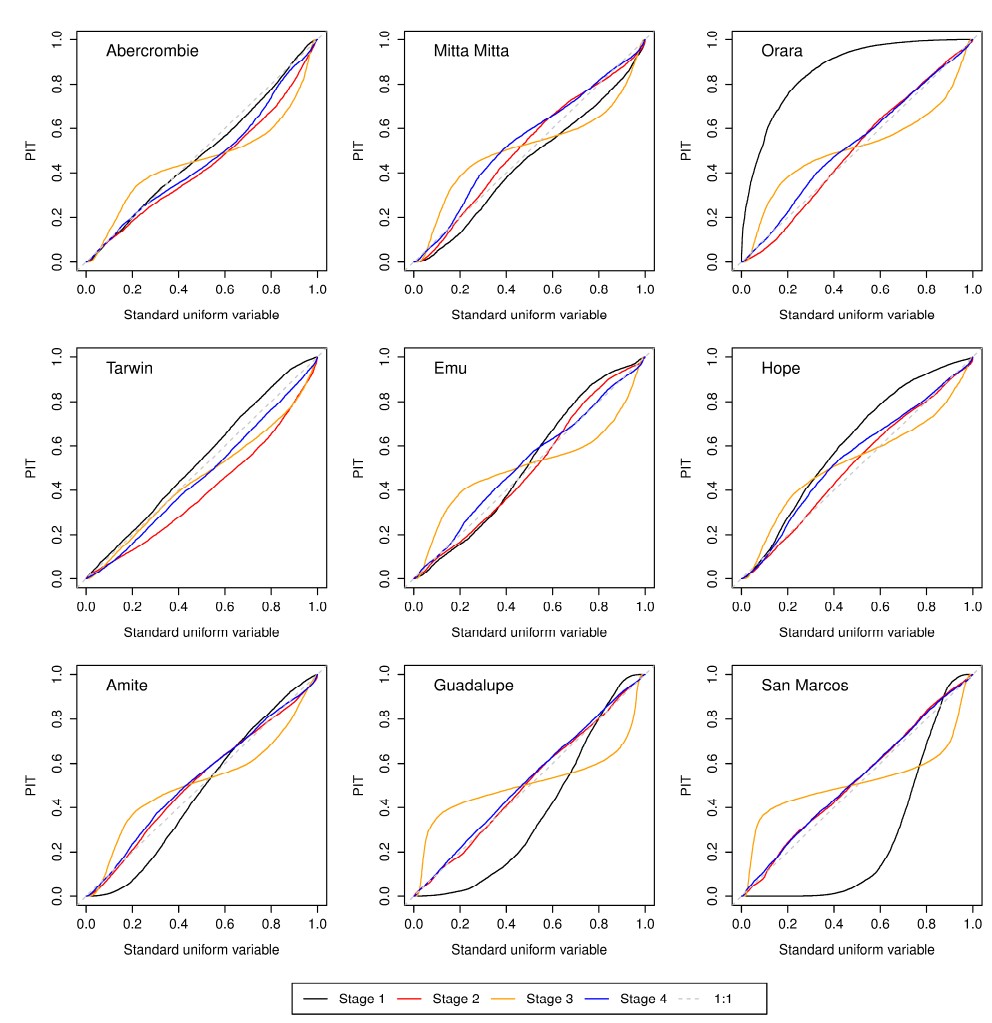

**Figure 9: Same as Figure 4 but the hydrological model is calibrated by the least-squares method.**



**Table 1: Summary of the ERRIS method**

|  | *Stage 1* | *Stage 2* | *Stage 3* | *Stage 4* |
|---|---|---|---|---|
| Purpose | Transformation and Hydrological model simulation | Linear bias correction | AR updating | Residual distribution refinement |
| Calibrated parameters | Hydrological model parameters, transformation parameters | bias-correction parameter | AR parameters | Distribution parameters |
| Correlation structure | Independent | Independent | Auto-correlated with lag one | Auto-correlated with lag one |
| Residual distribution | Transformed-Gaussian | Transformed -Gaussian | Transformed-Gaussian | Transformed- Gaussian mixture |






**Table 2: Basic catchment characteristics (1992-2005)**

| Name | Country | Gauge Site | Area (km$^2$) | Rainfall (mm/yr) | Streamflow (mm/yr) | Runoff coefficient | Zero flows |
|---|---|---|---|---|---|---|---|
| Abercrombie | Aus | Abercrombie River at Hadley no. 2 | 1447 | 783 | 63 | 0.08 | 14.4% |
| Mitta Mitta | Aus | Mitta Mitta River at Hinnomunjie | 1527 | 1283 | 261 | 0.20 | 0 |
| Orara | Aus | Orara River at Bawden Bridge | 1868 | 1176 | 243 | 0.21 | 0.6% |
| Tarwin | Aus | Tarwin River at Meeniyan | 1066 | 1042 | 202 | 0.19 | 0 |
| Emu | Aus | Mount Emu Creek at Skipton | 1204 | 641 | 23 | 0.04 | 0 |
| Hope | Aus | Mount Hope Creek at Mitiamo | 1646 | 436 | 11 | 0.02 | 23.3% |
| Amite | US | 07378500 | 3315 | 1575 | 554 | 0.35 | 0 |
| Guadalupe | US | 08167500 | 3406 | 772 | 104 | 0.13 | 1.7% |
| San Marcos | US | 08172000 | 2170 | 844 | 165 | 0.20 | 0 |







**Table 3: AWCI and CRPS calculated from the reference forecast for each catchment**

| | Abercrombie | Mitta Mitta | Emu | Hope | Orara | Tarwin | Amite | Guadalupe | San Marcos |
|---|---|---|---|---|---|---|---|---|---|
| $AWCI_{REF}$ (m³/s) | 18.00 | 49.68 | 9.41 | 5.04 | 62.83 | 38.81 | 409.63 | 70.25 | 59.69 |
| $CRPS_{REF}$ (m³/s) | 2.20 | 6.42 | 0.79 | 0.46 | 10.25 | 4.65 | 41.69 | 9.29 | 7.64 |






**Table 4: The calibrated error model parameters for the selected catchments.**

| Stage | Parameter | Catchment | | | | | | | | |
|---|---|---|---|---|---|---|---|---|---|---|
| | | Abercrombie | Mitta Mitta | Emu | Hope | Orara | Tarwin | Amite | Guadalupe | San Marcos |
| 1 | $x_1$ | 551.26 | 1319.05 | 485.73 | 561.36 | 481.28 | 672.24 | 1279.63 | 763.15 | 906.72 |
| | $x_2$ | -0.41 | -3.13 | -3.22 | -0.06 | 0.49 | -2.20 | -2.59 | 0.92 | 1.66 |
| | $x_3$ | 7.94 | 65.63 | 12.40 | 1.10 | 28.71 | 20.24 | 44.67 | 23.67 | 39.93 |
| | $x_4$ | 12.29 | 9.39 | 25.86 | 89.21 | 20.33 | 27.54 | 15.59 | 8.80 | 11.76 |
| | $\log(a)$ | -10.55 | -9.70 | -14.95 | -11.80 | -9.08 | -11.55 | -21.48 | -10.38 | -23.75 |
| | $\log(b)$ | -9.46 | -9.49 | -7.51 | -8.68 | -9.01 | -9.35 | -9.95 | -9.89 | -9.89 |
| | $\sigma_1$ | 5298.92 | 5233.01 | 1790.99 | 4523.05 | 4490.65 | 5271.08 | 8885.27 | 8366.75 | 6843.48 |
| 2 | $c$ | 6997.90 | -14341.19 | -373.84 | 946.83 | -3153.26 | -3282.81 | 1117.29 | 24909.80 | 10653.89 |
| | $d$ | 1.06 | 0.85 | 0.98 | 1.02 | 0.95 | 0.96 | 1.01 | 1.16 | 1.07 |
| | $\sigma_2$ | 5290.04 | 4924.38 | 1789.96 | 4540.44 | 4468.17 | 5244.14 | 8884.12 | 8025.35 | 6767.15 |
| 3 | $\rho$ | 0.86 | 0.95 | 0.96 | 0.97 | 0.95 | 0.94 | 0.86 | 0.83 | 0.82 |
| | $\sigma_3$ | 3289.50 | 1765.58 | 592.12 | 1611.67 | 1656.96 | 2154.72 | 5155.51 | 4661.31 | 4058.23 |
| 4 | $w$ | 0.73 | 0.69 | 0.77 | 0.70 | 0.75 | 0.64 | 0.55 | 0.86 | 0.87 |
| | $s_1$ | 1006.22 | 492.91 | 186.56 | 792.99 | 558.05 | 678.15 | 1481.79 | 1417.63 | 1246.49 |
| | $s_2$ | 6238.76 | 3092.35 | 1192.76 | 2693.45 | 3159.56 | 3473.87 | 7487.62 | 9573.92 | 10673.07 |






**Table 5: The calibrated parameters when Student's t distribution is used to describe the residual distribution at Stage 4**

|  | Abercrombie | Mitta Mitta | Emu | Hope | Orara | Tarwin | Amite | Guadalupe | San Marcos |
|---|---|---|---|---|---|---|---|---|---|
| $r$ | 1058.36 | 487.30 | 163.52 | 875.77 | 547.63 | 824.62 | 2033.78 | 1148.71 | 836.18 |
| $v$ | 1.44 | 1.25 | 1.33 | 2.31 | 1.53 | 1.58 | 1.62 | 1.36 | 1.54 |
