# Peer review of "Error reduction and representation in stages (ERRIS) in"

_Hydrology and Earth System Sciences, 2015_

## Referee Comment (RC1) · Anonymous Referee #1 · 2 Mar 2016

The paper is generally well written and well structured, which deserves to be published after some minor revisions. The presented work is an interesting extension of previous published papers from the authors and could become important for hydrologist working on operational forecast systems. Nonetheless there are a few points which should be addressed and clarified in more detail. The authors propose an interesting approach for modelling the forecast errors separated into four stages. Unfortunately the application is restricted to one step ahead predictions so far, which questions a bit the effort of implementing four different stages, since operational forecasts will need methods for longer lead times anyhow. Therefore it should be mentioned already in the abstract that the proposed method will be a theoretical exercise. Furthermore it should be stressed

right from the beginning that this paper represents some further developments of previous work of the authors, where they have published already parts of this work (stage 1 and stage 3), see Li (2015)! The term "ensemble" used within the paper is somewhat divergent to its usage in hydro-meteorology and I would suggest to use probabilistic or density forecasts instead of ensemble forecast. As far as I have understood you refer to "ensemble forecasts" because of the Gaussian distributed residuals in stages one to three (four). You should clarify how you understand ensembles, since your approach is not based on meteorological ensembles (e.g. ENS from ECMWF), which are usually the driving forces for creating hydrological ensembles (see HEPEX, which you have mentioned in the introduction). I find it also a bit surprising that the work of Krzysztofowicz, R. has not been mentioned, since there are quite a lot of analogies. Although his Hydrological Uncertainty Processor relies on Bayesian Theory, the different stages of ERRIS are implemented similarly: Transformation to the normal space (normal quantile mapping), bias correction (regression between observed and simulated series) and an Autoregressive AR(1) model. It would be good to cite some papers of Krzysztofowicz (1990,2001) and maybe follow up work (Todini, 2008; Reggiani, et a., 2009) and highlight the differences. Regarding the PIT diagrams: It should be mentioned that the usual analysing tool is the rank histogram (Hammil, 2001; Gneiting, et al., 2005), which is closely related to the diagrams used in this paper, which are, however, called "predictive quantile quantile" plots (Renard, 2010). Furthermore a Kolmogorov significant band should be included in these QQ diagrams as a test of uniformity (Laio and Tamea, 2007 (which is cited within the text, however missing in the reference list)). Since you use the CRPS for verification, it would make sense to decompose the CRPS into a reliability part and a resolution/uncertainty part (Hersbach, 2000). Thus the uncertainty part could be related to the average spread within the ensemble and the behaviour of its outliers, which would be an important information complementing the results of the $\alpha$–index and the AWCI and maybe confirm your interpretation of the decreased reliability of stage 3.

Some more technical comments:

Page 3, line 53: I don't agree with the statement that the aim of the ensemble is the reduction of the uncertainty! This is rather the aim of the post-processing of the ensembles.

Page 5, line 92: Since there are a lot of papers applying the Normal Quantile Transform, you should cite this paper at least: Krzysztofowicz, R. (1997)

Line 108-109: The forecast quality will more and more depend on the quality of the meteorological forecast and will dominate the uncertainty with increasing lead-time! (the same remark is valid for page 25, line 500)

Page 8, line 153: you assume a constant variance for the residuals in the normal space. In Figure 2 (a) it seems that the variance is varying depending on stream-flow. So this variability of the variance will stem from the back-transformation, I assume. Could you please clarify this.

Page 12, line 241: weights p = $\omega$, $\omega$-1

Page 14, line 269: Why are these parameters needed for estimation purposes only. That should be clarified.

Page 16, line 295 – 300: No references are given for sharpness, AWCI ! (E.g. Gneiting, et al., 2007 )

Line 306-308: I assume that you will need a reference forecast for each day (day/month/year) and not only one per month (month/year) and that you will take the mean of the 1000 samples per day? Could you please clarify this?

Page 18: Here you mention the GR4J model, in table 4 you show 4 hydrological model parameters x1,..x4, could you please give more details about the model and explain the meaning of these four parameters

Refrences:

Hamill, T. M. (2001) Interpretation of rank histograms for verifying ensemble forecasts.

Monthly Weath. Rev., 129, 550–560.

Gneiting, T., Raftery, A. E., Westveld, A. H. and Goldman, T. (2005) Calibrated probabilistic forecasting using ensemble model output statistics and minimum CRPS estimation. Monthly Weath. Rev., 133, 1098–1198.

Gneiting, T., Balabdaoui, F.and Raftery, A., 2007. Probabilistic forecasts, calibration and sharpness. Journal of the Royal Statistical Society. Series B: Statistical Methodology 69, 243–268.

Hersbach H. 2000. Decomposition of the continuous ranked probability score for ensemble prediction systems. Weather and Forecasting 15: 559–570. Krzysztofowicz, R. (1997), Transformation and normalization of variates with specified distributions, J. Hydrol., 197(1–4), 286–292.

Krzysztofowicz, R. (1999), Bayesian theory of probabilistic forecasting via deterministic hydrologic model, Water Resour. Res., 35(9),2739–2750.

Krzysztofowicz, R. (2001), The case for probabilistic forecasting in hydrology, J. Hydrol., 249(1–4), 2-9.

Laio, F., and S. Tamea (2007), Verification tools for probabilistic forecasts of continuous hydrological variables, Hydrol. Earth Syst. Sci., 11(4), 1267–1277.

Todini, E. (2008), A model conditional processor to assess predictive uncertainty in flood forecasting, Int. J. River Basin Manage., 6(2), 123–137.

Reggiani, P., Renner, M., Weerts, A., and van Gelder, P.: Uncertainty assessment via Bayesian revision of ensemble streamflow predictions in the operational river Rhine forecasting system, Water Resources Research, 45, 2009.

---

## Referee Comment (RC2) · Anonymous Referee #2 · 8 May 2016

It is a very well written paper, with an excellent overview. Figures illustrate the text well. The paper is mathematically rigorous. A considerable number of case studies are used for validation. Conclusions are well justified. Can be recommended for publication after some revisions; please see below.

The components of the approach are not new (distribution transforms and linear bias corrections and AR) but their combination seems to be useful, and their sequence is quite well justified. With respect to earlier literature, there are a couple of references which would be good to include, and to discuss the similarity and differences of these approaches.

1) Kelly and Krzysztofowicz, 1997; Montanari and Brath 2004 – meta-Gaussian approach.

2) Solomatine and Shrestha (2009) presented a method of predicting residual error distribution. They make no assumptions about the error distributions (which is perhaps a weakness) but simply build a non-linear regression model (neural network) model able to predict quantiles of this distribution at each time step. This model is not autoregressive but uses more information about the state of the system.

3) In their DUMBRAE method Pianosi and Raso (2012) use an approach similar to what is presented in the reviewed paper – a residual error corrector and AR model. It would be very useful to compare the presented approach to DUMBRAE.

—

I would make it clearer when you talk about a single model, and when about an ensemble. It is not immediately clear what the "error model" is – is a model providing the residual error, or the residual error distribution?

On page 9 it is mentioned that the flow forecast is the median of the ensemble flow forecast. This effectively makes model Qwave a deterministic model. It would be useful to clarify: is the presented method to be used only with the models using ensemble rainfall forecasts, or it can be applied to any (deterministic) model with a single (non-ensemble) input?

I suggest to provide a reference to Table 1 somewhat earlier – I think this would make reading easier.

–References–

Kelly, K. S., and R. Krzysztofowicz (1997), A bivariate meta-Gaussian density for use in hydrology, Stochastic Hydrol. Hydraul., 11(1), 17– 31, doi:10.1007/BF02428423.

Montanari, A., and A. Brath (2004), A stochastic approach for assessing the uncertainty of rainfall-runoff simulations, Water Resour. Res., 40, W01106,

doi:10.1029/2003WR002540.

D.P. Solomatine, D.L. Shrestha (2009). A novel method to estimate model uncertainty using machine learning techniques. Water Resources Res. 45, W00B11, doi:10.1029/2008WR006839.

F. Pianosi and L. Raso (2012). Dynamic modeling of predictive uncertainty by regression on absolute errors. Water Resources Research, 48, W03516, doi:10.1029/2011WR010603.

---

## Author Comment (AC1) · 28 May 2016

Thank you very much for your insightful and constructive comments and suggestions. We have carefully revised the manuscript to address all your concerns. We attach our specific response to each of your comments and the revised manuscript in the attachment for your consideration.

Please also note the supplement to this comment: http://www.hydrol-earth-syst-sci-discuss.net/hess-2015-514/hess-2015-514-AC1-supplement.zip

---

## Editor Comment (EC1) · D. Solomatine (Editor) · 29 Jun 2016

A well written paper, with the presentation of a version of an error correcting scheme (includes distributions transformations and AR). The components of it are not new but the combination has been shown to be quite workable and leading to model error reduction. Referees have suggested some clarifications, and to add a comparison to similar appraches publihsed earlier. The authors have done this, and the paper will be sent to referees for further review and to seek their recommendation for publication in HESS (and perhaps further suggestions).

---

## Author Response (AR1)

The paper is generally well written and well structured, which deserves to be published after some minor revisions. The presented work is an interesting extension of previous published papers from the authors and could become important for hydrologist working on operational forecast systems. Nonetheless there are a few points which should be addressed and clarified in more detail.

*Reply: Thank you very much for your insightful and constructive comments and suggestions. We have carefully revised the manuscript to address all your concerns. We attach our specific response to each of your comments below.*

The authors propose an interesting approach for modelling the forecast errors separated into four stages. Unfortunately the application is restricted to one step ahead predictions so far, which questions a bit the effort of implementing four different stages, since operational forecasts will need methods for longer lead times anyhow. Therefore it should be mentioned already in the abstract that the proposed method will be a theoretical exercise.

*Reply: The AR(1) model used in Stage 2 directly lead to one step ahead predictions, but ERRIS can be extended to multiple step ahead predictions by using AR(1) updating iteratively. We focus this work on some foundational issues of ERRIS, such as model structure and model inference. The model performance of direct one-step ahead forecasts is used to check model assumptions and understand the benefits of staged error modelling approaches. ERRIS is supposed to be an important component of future operational forecasting systems in Australia.*

*We agree that we should make it clear in the abstract that this paper is only tested the application of one-step ahead forecasts. We have amended the abstract as follows.*

> **Page 3 Line 53:**
>
> In a case study, we apply ERRIS for one-step ahead forecasting at a range of catchments.

Furthermore it should be stressed right from the beginning that this paper represents some further developments of previous work of the authors, where they have published already parts of this work (stage 1 and stage 3), see Li (2015)!

*Reply: We have followed your suggestions and added the connections to our previous work.*

> **Page 6 Line 124:**
>
> ERRIS is a further development of the restricted autoregressive model [*Li et al.*, 2015] and a seasonal error model developed by *Li et al. [2013]*.

The term "ensemble" used within the paper is somewhat divergent to its usage in hydro-meteorology and I would suggest to use probabilistic or density forecasts instead of ensemble forecast. As far as I have understood you refer to "ensemble forecasts" because of the Gaussian distributed residuals in stages one to three (four). You should clarify how you understand ensembles, since your approach is not based on meteorological ensembles (e.g. ENS from ECMWF), which are usually the driving forces for creating hydrological ensembles (see HEPEX, which you have mentioned in the introduction).

*Reply: We have added clarification of the meaning of "ensemble" in this work.*

> **Page 7 Line 135:**

> In this study we use the term "ensemble" to mean a set of equally probable realizations of future streamflow that represents the hydrological model uncertainty. The forecasts based on ERRIS are not typical probabilistic forecasts [*Gneiting and Katzfuss*, 2014], which explicitly provide the predictive distribution of future streamflow. For ERRIS, the probability distribution may be theoretically derived for one-step ahead forecasts based on the distributional assumption of model residuals. However, we can only obtain the predictive distribution of ERRIS forecasts at multiple step by means of Monte Carlo simulation.

I find it also a bit surprising that the work of Krzysztofowicz, R. has not been mentioned, since there are quite a lot of analogies. Although his Hydrological Uncertainty Processor relies on Bayesian Theory, the different stages of ERRIS are implemented similarly: Transformation to the normal space (normal quantile mapping), bias correction (regression between observed and simulated series) and an Autoregressive AR(1) model. It would be good to cite some papers of Krzysztofowicz (1990,2001) and maybe follow up work (Todini, 2008; Reggiani, et a., 2009) and highlight the differences.

*Reply: It is very interesting to know that ERRIS shares some elements with HUP, though two methods were designed independently. We have added a paragraph to compare ERRIS with HUP.*

**Page 28 Line 556:**

> The staged approach of ERRIS sets it apart from several predecessors, for example the hydrological uncertainty processor (HUP) and the dynamic uncertainty model by regression on absolute error (DUMBRAE). HUP is a Bayesian forecasting system to produce probabilistic streamflow forecasts [*Kelly and Krzysztofowicz*, 1997; *Krzysztofowicz*, 1999; 2001; *Krzysztofowicz and Kelly*, 2000; *Reggiani et al.*, 2009; *Todini*, 2008]. HUP and ERRIS have some similarities: (1) both are post-processors of deterministic hydrological models for hydrological uncertainty quantification; (2) both apply transformation to normalize data; (3) both use a linear regression in the transformed space for bias correction; (4) both use an autoregressive model to update hydrological simulation. However, ERRIS differs fundamentally from HUP by being implemented in stages. As we have noted, the staged approach avoids unwanted the interaction between parameters, and ensure the base hydrological model performs as strongly as possible. In addition, some other technical advances distinguish ERRIS from HUP. For instance, ERRIS applies a restricted autoregressive model in order to avoid the possible overcorrection from the ordinary autoregressive model used in HUP. ERRIS uses a mixture of two Gaussian distributions for the residual distribution, which is more flexible than a Gaussian distribution used in HUP to describe the peak, shoulder and tail of the distribution.

Regarding the PIT diagrams: It should be mentioned that the usual analysing tool is the rank histogram (Hammil, 2001; Gneiting, et al., 2005), which is closely related to the diagrams used in this paper, which are, however, called "predictive quantile quantile" plots (Renard, 2010). Furthermore a Kolmogorov significant band should be included in these QQ diagrams as a test of uniformity (Laio and Tamea, 2007 (which is cited within the text, however missing in the reference list)).

*Reply: We have added more descriptions of the PIT plots and include the Kolmogorov significant band in Figures 4 and 9.*

**Page 16 Line 303**

We graphically examine $\pi_t$ with the corresponding theoretical quantile of the uniform distribution using the PIT-uniform probability plot (or simply *PIT plot;* also called the predictive quantile quantile

plot [*Renard et al.*, 2010]). The PIT plot is closely related to the rank histogram [*Gneiting et al.*, 2005; *Hamill*, 2001]. From our experience, the PIT plot is more suitable than the rank histogram for the experiments where observations are abundant (such as daily or sub-daily forecasting verification). A perfectly reliable forecast follows the 1:1 line. A Kolmogorov-Smirnov significant band can be included in the PIT plots to as a test of uniformity [*Laio and Tamea*, 2007].

**Page 21 Line 415**

As PIT values are highly autocorrelated, we have to "thin" them in order to make the Kolmogorov-Smirnov significant band applicable [*Zhao et al.*, 2015]. We generate PIT plots from every 30-th forecast to eliminate the autocorrelation.

[Figure]

**Figure 4: Comparison of the cumulative probability distribution of the PIT at different stages (light blue shaded strips indicate the 95% significant band of the Kolmogorov-Smirnov test.)**

[Figure]

**Figure 9: Same as Figure but the hydrological model is calibrated by the least-squares method.**

Since you use the CRPS for verification, it would make sense to decompose the CRPS into a reliability part and a resolution/uncertainty part (Hersbach, 2000). Thus the uncertainty part could be related to the average spread within the ensemble and the behaviour of its outliers, which would be an important information complementing the results of the α–index and the AWCI and maybe confirm your interpretation of the decreased reliability of stage 3.

*Reply: This is an interesting suggestion. We show the CRPS decomposition in the Table 1, below. We found that the reliability part does not always support the improvement of reliability at Stage 4 as suggested by the PIT plots.*

*This is caused by the fact that the CRPS decomposition interprets the reliability differently from the PIT plots (or equivalently the rank histogram). Hersbach (2000) stated that "the reliability of CRPS is sensitive to the width of the ensemble bins, while the rank histogram gives each forecast the same weight". The CRPS decomposition by Hersbach (2000) actually defines a weaker condition of reliability [Candille and Talagrand, 2005]. The reliability part has a dimension that is related to the ensemble spread. Therefore, the reliability part actually measures both reliability and sharpness. The use of a mixture of two Gaussian distributions at Stage 4 effectively corrects the shape of the predictive distribution, especially the peak and shoulders of the distribution. Nevertheless, Stage 4 sometimes produces longer tailed predictive distribution than Stage 3 and has to sacrifice the forecast sharpness as a result.*

*We have added clarification on the different interpretation between the reliability part and the PIT plots. In order to avoid the confusion of different interpretations of reliability from the CRPS decomposition and the PIT plots, we chose not to report Table 1.*

**Table 1: The reliability part in the CRPS decomposition**

|  | Stage 1 | Stage 2 | Stage 3 | Stage 4 |
|---|---|---|---|---|
| Abercrombie | 0.028 | 0.012 | 0.034 | 0.056 |
| Mitta Mitta | 0.203 | 0.010 | 0.028 | 0.010 |
| Orara | 0.061 | 0.061 | 0.007 | 0.009 |
| Tarwin | 0.001 | 0.003 | 0.001 | 0.001 |
| Emu | 0.103 | 0.035 | 0.039 | 0.078 |
| Hope | 0.021 | 0.043 | 0.013 | 0.038 |
| Amite | 0.580 | 0.576 | 0.337 | 0.619 |
| Guadalupe | 0.188 | 0.034 | 0.112 | 0.139 |
| San Marcos | 0.076 | 0.038 | 0.091 | 0.090 |

**Page 18 Line 347**

While a decomposition of CRPS is available that gives an indication of reliability [*Hersbach 2000*], we do not use this. PIT plots are a stronger test of reliability [*Candille and Talagrand*, 2005], and accordingly we focus on PIT plots to discuss reliability.

Some more technical comments:

Paper Page 3, line 53: I don't agree with the statement that the aim of the ensemble is the reduction of the uncertainty! This is rather the aim of the post-processing of the ensembles.

*Reply: Thanks for your comment, but we respectfully disagree. We believe it is generally true that ensemble forecast researchers in all fields aim make their forecasts as accurate and sharp as possible. This implies reducing the uncertainty as much as possible. Indeed, many meteorological forecasts are over-confident, and post-processors are required to make the forecasts more (not less) uncertain.*

Page 5, line 92: Since there are a lot of papers applying the Normal Quantile Transform, you should cite this paper at least: Krzysztofowicz, R. (1997)

*Reply: We have followed your suggestion:*
* * *
**Page 4 Line 89**

Heteroscedasticity and non-Gaussianity of residuals may also be dealt with implicitly, and often more conveniently, by using data transformation to normalize the residuals and stabilize their variance, such as the normal quantile transform [*Kelly and Krzysztofowicz*, 1997; *Krzysztofowicz*, 1997; *Montanari and Brath*, 2004], the Box-Cox transformation [*Thyer et al.*, 2002] and the log-sinh transformation [*Wang et al.*, 2012].
* * *
Line 108-109: The forecast quality will more and more depend on the quality of the meteorological forecast and will dominate the uncertainty with increasing lead-time! (the same remark is valid for page 25, line 500)

*Reply: We restrict our study on the hydrological model uncertainty and do not consider the uncertainty of meteorological forcing variables. I agree that the quality of the meteorological forecasts will be a dominant factor to forecast uncertainty at longer lead times. We clarify this point more clearly in the revision.*
* * *
**Page 6 Line 116**

Thus as lead-time increases, and the influence of the error model decreases, the quality of the forecast relies on the performance of the base hydrological model and the quality of meteorological forecasts [*Bennett et al.*, 2014a].
* * ** * *
**Page 26 Line 529**

Thus forecasts at long lead times rely strongly on the hydrological model (and, in our case, a bias-correction) to be plausible, even with perfect meteorological forcings.
* * *
Page 8, line 153: you assume a constant variance for the residuals in the normal space. In Figure 2 (a) it seems that the variance is varying depending on stream-flow. So this variability of the variance will stem from the back-transformation, I assume. Could you please clarify this.

*Reply: You are absolutely correct. A constant variance in the transformed space becomes a varying variance in the original space through the back-transformation. We make this clear in the revision.*
* * *
**Page 8 Line 166**

The forecast variance in the original (untransformed) space is not a constant but is dependent on the magnitude of simulated streamflow through the back-transformation.
* * *
Page 12, line 241: weights $p = \omega, \omega-1$

*Reply: We have changed the notation.*

**Page 13 Line 254**

where $MN\left(\tilde{Z}_4(t), \sigma_{4,1}^2, \sigma_{4,2}^2, w\right)$ represents a mixture of two Gaussian distributions $N\left(\tilde{Z}_4(t), \sigma_{4,1}^2\right)$ and $N\left(\tilde{Z}_4(t), \sigma_{4,2}^2\right)$ with weights $w$ and $1-w$.

Page 14, line 269: Why are these parameters needed for estimation purposes only. That should be clarified.

*Reply: We have added some clarification as follows.*

**Page 15 Line 282**

The variances at Stages 1-3 (i.e. $\sigma_1$, $\sigma_2$ and $\sigma_3$) are not used to generate forecasts, but only for estimating parameters at corresponding stages. We use maximum likelihood at each stage to estimate parameters, and this requires us to specify the variance of residuals at each stage.

Page 16, line 295 – 300: No references are given for sharpness, AWCI ! (E.g. Gneiting, et al., 2007 ) Line 306-308: I assume that you will need a reference forecast for each day (day/month/year) and not only one per month (month/year) and that you will take the mean of the 1000 samples per day? Could you please clarify this?

*Reply: We have added the reference for AWCI. Considering only 14-year data are used, the reference forecast for each day (day/month/year) is duplicated 14 values at most and is not robust based on our experience. In order to increase the robustness of the reference forecast, we chose to the reference forecast for each month. We have clarified this point in the revision.*

**Page 17 Line 322**

We use the average width of the 95% forecast intervals (AWCI) to indicate forecast sharpness [*Gneiting et al.*, 2007].

**Page 17 Line 341**

As only 14years of data are used in this study, the reference forecast for each month is more robust than the similar reference forecast for each day.

Page 18: Here you mention the GR4J model, in table 4 you show 4 hydrological model parameters x1,..x4, could you please give more details about the model and explain the meaning of these four parameters

*Reply: We have added the explanation of GR4J parameters.*

**Page 19 Line 364**

GR4J is a widely used conceptual model that was designed to be as parsimonious as possible. Its four parameters describe the depth of a production store (X1), groundwater exchange (X2), the depth of a routing store (X3) and the length of unit hydrographs (X4).

formulates the error variance as a function of time series of absolute hydrological model errors and several independent predictors (such as precipitation). The dynamic variance model of DUMBRAE is an interesting alternative to the method we have presented here. As with HUP, another major difference between ERRIS and DUMBARE is staged error modelling that allows ERRIS to characterize the forecast error in stages and to avoid potential parameter interference and ensure robust performance of the base hydrological model.

I would make it clearer when you talk about a single model, and when about an ensemble. It is not immediately clear what the "error model" is – is a model providing the residual error, or the residual error distribution?

*Reply: We are not really sure whether we understand this comment correctly. It seems to relate to another comment from Reviewer #1 about the term "ensemble". We have added clarification on the meaning of "ensemble" in the context of this study.*

**Page 7 Line 135:**

In this study we use the term "ensemble" to mean a set of equally probable realizations of future streamflow that represents the hydrological model uncertainty. The forecasts based on ERRIS are not typical probabilistic forecasts [*Gneiting and Katzfuss*, 2014], which explicitly provide the predictive distribution of future streamflow. For ERRIS, the probability distribution may be theoretically derived for one-step ahead forecasts based on the distributional assumption of model residuals. However, we can only obtain the predictive distribution of ERRIS forecasts at multiple step by means of Monte Carlo simulation.

On page 9 it is mentioned that the flow forecast is the median of the ensemble flow forecast. This effectively makes model Qwave a deterministic model. It would be useful to clarify: is the presented method to be used only with the models using ensemble rainfall forecasts, or it can be applied to any (deterministic) model with a single (nonensemble) input?

*Reply: We apologise for this point of confusion. When we discuss the 'median', what we are stating is that, in the original space, we assume that the streamflow simulation is the median of the distribution of the residuals. (After transformation, the median and the mean should be approximately equal.) The streamflow simulation (or forecast) is always deterministic. We are not describing the median resulting from an ensemble of rainfall forecasts.*

*There are two points of clarification that we hope will address the reviewer's query about how ERRIS will interact with meteorological ensemble forecasts:*

*1) ERRIS is designed to be applied only to a single streamflow realisation*

*2) ERRIS does not account for uncertainty in rainfall forecasts*

*To use ERRIS with a meteorological ensemble, each member of the rainfall ensemble would be run through GR4J, and ERRIS would be applied to each ensemble trace. If ERRIS is applied to a deterministic meteorological forecast, the uncertainty of the rainfall forecast would not be present in the streamflow forecast, and the forecasts would not be reliable. This is particularly important at longer lead times, where rainfall forecast uncertainty tends to be the most significant contributor to overall streamflow forecast uncertainty. We designed ERRIS to be used in conjunction with ensemble meteorological forecasts (specifically, the method described by Robertson et al. 2013 and Shrestha et al. 2015). This allows us to separately quantify of rainfall and hydrological uncertainties – a significant benefit of the method.*

*We rephrase our arguments as following*

> **Page 6 Line 127**
>
> We assume that errors in streamflow forecasts due to weather forecasts (precipitation in particular) will be considered separately by using ensemble weather forecasts [*Bennett et al.*, 2014b; *Robertson et al.*, 2013; *Shrestha et al.*, 2013], and we do not consider these in this paper.

I suggest to provide a reference to Table 1 somewhat earlier – I think this would make reading easier.

***Reply****: We have followed your suggestion and moved Table 1 earlier.*

> **Page 10 Line 196**
>
> Stage 1 of the ERRIS method is summarized in Table 1.

**References**

[revised manuscript text omitted]